# Bacteriophages as Potential Tools for Use in Antimicrobial Therapy and Vaccine Development

**DOI:** 10.3390/ph14040331

**Published:** 2021-04-05

**Authors:** Beata Zalewska-Piątek, Rafał Piątek

**Affiliations:** 1Department of Molecular Biotechnology and Microbiology, Chemical Faculty, Gdańsk University of Technology, Narutowicza 11/12, 80-233 Gdańsk, Poland; rafpiate@pg.edu.pl; 2BioTechMed Center, Gdańsk University of Technology, Narutowicza 11/12, 80-233 Gdańsk, Poland

**Keywords:** phages, phage-based vaccines, phage display technology, phage DNA vaccines, bacterial resistance, phage therapy

## Abstract

The constantly growing number of people suffering from bacterial, viral, or fungal infections, parasitic diseases, and cancers prompts the search for innovative methods of disease prevention and treatment, especially based on vaccines and targeted therapy. An additional problem is the global threat to humanity resulting from the increasing resistance of bacteria to commonly used antibiotics. Conventional vaccines based on bacteria or viruses are common and are generally effective in preventing and controlling various infectious diseases in humans. However, there are problems with the stability of these vaccines, their transport, targeted delivery, safe use, and side effects. In this context, experimental phage therapy based on viruses replicating in bacterial cells currently offers a chance for a breakthrough in the treatment of bacterial infections. Phages are not infectious and pathogenic to eukaryotic cells and do not cause diseases in human body. Furthermore, bacterial viruses are sufficient immuno-stimulators with potential adjuvant abilities, easy to transport, and store. They can also be produced on a large scale with cost reduction. In recent years, they have also provided an ideal platform for the design and production of phage-based vaccines to induce protective host immune responses. The most promising in this group are phage-displayed vaccines, allowing for the display of immunogenic peptides or proteins on the phage surfaces, or phage DNA vaccines responsible for expression of target genes (encoding protective antigens) incorporated into the phage genome. Phage vaccines inducing the production of specific antibodies may in the future protect us against infectious diseases and constitute an effective immune tool to fight cancer. Moreover, personalized phage therapy can represent the greatest medical achievement that saves lives. This review demonstrates the latest advances and developments in the use of phage vaccines to prevent human infectious diseases; phage-based therapy, including clinical trials; and personalized treatment adapted to the patient’s needs and the type of bacterial infection. It highlights the advantages and disadvantages of experimental phage therapy and, at the same time, indicates its great potential in the treatment of various diseases, especially those resistant to commonly used antibiotics. All the analyses performed look at the rich history and development of phage therapy over the past 100 years.

## 1. Introduction

The increasing number of antibiotic-resistant bacteria correlates with increased healthcare costs, morbidity, and mortality worldwide. The World Health Organization (WHO, Geneva, Switzerland) correctly notes that antibiotic resistance is one of the greatest threats to public health and global development. In 2016, WHO prepared the tenth revision of the international statistical classification of diseases and related health problems, distinguishing 17 disease categories (Appendix A) [1]. In addition, WHO unveiled a global list of antibiotic-resistant bacteria to intensify the research and development of new drugs against resistant bacteria. The analyses showed that 60% of pathogenic bacteria have become resistant to commonly used antibiotics, including carbapenems and third-generation cephalosporins (Appendix A) [2,3].

An alternative solution to the growing resistance of bacteria to antibiotics can be phage therapy, based on lytic phages or a combination of phages and antibiotics [4,5,6]. This antibacterial technology has been known for 100 years and can be useful not only in combating antibiotic-resistant and tolerant bacteria but also in treating infections related to biofilm formation as well as against spore formers [7,8,9,10,11].

Recent data also indicate the effectiveness of personalized phage therapy in some individuals in treating a variety of resistant bacterial infections, e.g., caused by *Escherichia coli*, *Staphylococcus aureus*, and *Pseudomonas aeruginosa* responsible for wound infections, multidrug-resistant (MDR) strains of *Pseudomonas aeruginosa* in cystic fibrosis (CF) patients, and uropathogenic bacteria (i.e., *E. coli*, *Streptococcus* spp., *P. aeruginosa*, *Proteus mirabilis*, *Staphylococcus aureus* causing urinary tract infections (UTIs)), or opportunistic nosocomial (e.g., *Acinetobacter baumannii*) leading to meningitis, pneumonia, and wound site and biofilm-related infections [12,13,14,15]. Phages can also be effective in fighting infections in patients who have undergone solid organ transplantation, such as lung transplant recipients suffering from multidrug-resistant *P. aeruginosa* or CF patients with disseminated drug-resistant *Mycobacterium abscessus* [16,17].

Moreover, phage vaccines in the form of phage-displayed vaccines or phage DNA vaccines are currently of great interest among various research groups [18,19,20,21,22]. These types of vaccines can become important in preventing bacterial infections and the diseases resulting from them (especially those caused by MDR strains). Phage vaccines have a number of properties that make them superior to traditional vaccines and enable them to overcome their limitations [18,19]. Conventional attenuated and inactivated live vaccines are widely used all over the world to prevent various human infectious diseases (mainly of bacterial and viral origin) by eliciting a protective immune response to specific antigens. The efficiency of this kind of pharmaceutical preparations is generally high. Nevertheless, there are some problems related to their transport, storage, and targeted delivery, as well as designing more effective immunogenic preparations. In addition, the side effects and safety of such vaccines can also be a serious concern, hence the growing need to design and produce new vaccines, e.g., based on phages. These vaccines are safe for immunized persons because bacterial viruses are not infectious agents for eukaryotes and do not cause pathogenic effects in humans [20,21]. Moreover, the vaccine preparations are chemically stable, cheap to produce, and easy to transport and store. They also exhibit immunostimulating and adjuvant properties [22]. Therefore, phage vaccines can be a great alternative for future vaccine development.

Methods for isolating, storing, and creating phage depositories are now becoming more available and better developed. The ATCC (American Type Culture Collection) and Public Health England (PHE) collections (including the National Collection of Type Cultures (NCTC)) should be distinguished among the largest and best-known phage resources. The above phage banks contain various phages that target pathogenic bacteria and therefore can be used for research and clinical purposes. One recent report provides information about an Israeli bank of 300 phages targeting 16 pathogenic bacterial species, along with a protocol used to isolate and characterize new phages [23]. For this reason, research on phages, their preparation for treatment, and the design of innovative vaccines based on them can be intensified. Studies of this type could be crucial in the face of the global emergence of antibiotic-resistant bacteria and the treatment of diseases caused by them.

## 2. Selected Aspects of the History and Development of Phage Therapy

Bacteriophages are viruses that infect bacterial cells and are widely distributed in the environment. The history of the discovery of phages and their subsequent use in the treatment of infectious diseases caused by resistant bacterial strains is extensive (Table 1). The first reports of the occurrence of bacterial parasites in the environment date back to 1896. They concerned the study of the waters of the Ganges and Jamuna Rivers in India by Ernest Hankin. This English bacteriologist did not discover phages but showed the presence of an unidentified substance (in the river water samples) with antibacterial activity against *Vibrio cholerae* [24,25]. Two years later, a similar phenomenon was observed by the Russian bacteriologist Nikolay Gamaleya while working with *Bacillus subtilis* [26].

The official discoverer (1917) of phages is Felix d’Herrelle (French-Canadian microbiologist from the Pasteur Institute in Paris), who, unlike other researchers, continued his research on these bacterial viruses. Working independently of d’Herell, Frederick Twort (British bacteriologist) in 1915 hypothesized that ultramicroscopic viruses may be responsible for the phenomenon of antibacterial activity [27]. However, for various reasons, including financial difficulties, Twort did not pursue his discovery. Unlike Hankin and Twort, d’Herelle had no doubts about the nature of the observed bactericidal phenomenon and proposed that it was caused by a virus capable of parasitizing bacteria. Moreover, d’Herelle (October 18, 1916) also invented the term bacteriophages consisting of the two words bacteria and phagein [28,29,30]. However, the first d’Herelle research to discover phages and develop treatments for bacterial infections caused by *Shigella* strains in humans took place in July–August 1915 during the investigation of an outbreak of severe dysentery hemorrhage in French soldiers (Maisons-Laffitte, outskirts of Paris, France) [28,29].

Two years after the discovery of phages (in 1919), d’Herelle had the opportunity to try his antidisentric phage in the treatment of dysentery at the Hospital des Enfants-Malades in Paris. Before using phage therapy in children, its safety was checked after administering phages to the discoverer himself, the clinical supervisor Professor Victor-Henri Hutinel (the hospital’s head of pediatrics), and hospital interns. One dose of phage preparation was sufficient for health during 24 h of treatment [30].

In 1923, George Eliava (prominent Georgian bacteriologist), with the support of d’Herelle, founded the Eliava Institute of Bacteriophage, Microbiology, and Virology (EIBMV) of the Georgian Academy of Sciences in Tbilisi, which became a leading center for the production and use of therapeutic phages [25]. In 1935, d’Herelle wrote a book (in Russian) concerning phages and the process of human healing from bacterial infections [31]. In 1939m an article was published (as part of the *Archiv für die Gesamte Virusforschung*, currently *Archives of Virology*) emphasizing the importance of electron microcopy in the study and visualization of viruses. In this article, Dr. Helmut Ruska (Charité Medical School in Berlin) showed for the first time an electron microscope examination of bacterial lysis caused by phages [32]. Until then, none of the researchers had observed the phage particles. This was an important breakthrough in bacteriophage studies.

Soon after the success of dysentery treatment, commercial phage production began at the d’Herelle laboratory in Paris. Phage preparations (e.g., Bacte’-coli-phage, Bacte´-rhinophage, Bacte´-pyo-phage, Bacte´-intesti-phage, and Bacte´-staphy-phage) were introduced to the market by the French company L’Oreal. Then, the Indianapolis company of Eli Lilly (1940) produced phage preparations in both forms, as water-soluble gel preparations (e.g., Ento-jel, Colo-jel, and Staphylo-jel) and sterile phage lysates (e.g., Ento-lysate, Colo-lysate, Staphylo-lysate, Neiso-lysate) to treat infections caused by *E. coli*, *Staphylococcus* sp., *Streptococcus* sp. and other pathogenic strains [33,34]. The production of phages and their use in the treatment of infectious diseases in Europe, Asia, and America continued until the outbreak of World War II and the discovery and expansion of the host-range antibiotics (starting with penicillin in 1928). At the time, phage therapy was developing dynamically, and other pharmaceutical companies were producing phage preparations. After World War II, this method of treatment was gradually replaced by antibiotics that were easier to apply and more effective in the treatment of various infections affecting humanity. Despite the boom of antibiotics, phages were still used to fight bacterial pathogens in the former Soviet Union, Georgia, and Poland, while they were ignored in Western countries [25,35]. However, due to the increasing incidence of antibiotic-resistant bacterial strains (mainly in hospital settings), the role of phage therapy and the use of phages in vaccine production began to increase again in the 1980s. The first phase of phage research concerned the sequencing of the filamentous phage ΦX174 genome [36]. In 1985, the novel expression vectors based on the filamentous fusion phage (as the phage display technology) were invented [37]. In 1988, the idea of creating a phage display library was born [38]. Then, in 1989 and 1990, immunoglobulin fragments (including Fab and variable, V domains, respectively) were used by Lerner and Winter as fusion peptides for phage surface display (based on the novel λ vector system and the filamentous fd phage, respectively) [39,40]. Both of the above-mentioned scientists, in 2012, received the prestigious Prince of Asturias Award for Technical and Scientific Research (in recognition for their contributions to the field of immunochemistry and combinatorial antibodies). In 1988, a genetically engineered filamentous phage with circumsporozoite protein displayed on its surface was used as a model phage-displayed vaccine against malaria caused by *Plasmodium falciparum*. The foreign antigen was genetically fused to a minor coat protein by cloning the target gene into the pIII gene of the filamentous phage. Then, based on rabbit and mouse models, the recombinant phages were shown to be antigenic and immunogenic [41]. Finally, in 2018, research on phage display and the application of this method for the directed evolution of antibodies with the aim of producing new pharmaceuticals was awarded the Nobel Prize in Chemistry (shared/jointly by George P. Smith and Sir Gregory P. Winter) [18].

An important contribution of Polish scientists to the development of phage therapy was related to the establishment of the Hirszfeld Institute of Immunology and Experimental Therapy (HIIET) of the Polish Academy of Sciences (Wroclaw, Poland) in 1952. Phage therapy research in the institute began in 1957 by treating infections caused by *Shigella*. In addition, the bacteriophage laboratory of HIIET produced phages necessary for the treatment of pulmonary and urinary tract infections, septicemia, furunculosis, and post-traumatic postoperative infections (including prophylaxis). The institute’s researchers also developed (in English) the most detailed studies on the clinical application of phage preparations. Further research on phage therapy was carried out in the 1970s and 1980s by Prof. Stefan Ślopek (successor of Prof. L. Hirszfeld, the founder of HEIIT) [42,43,44,45,46,47]. In 1999, Prof. A. Górski became the director of the institute, who established a Phage Therapy Center (PTC) at HIIET. Currently, HIIET still continues experimental phage treatment of bacterial infections. The established PTC is an experimental phage therapy center operating in accordance with the current ethical, legislative, and administrative requirements in the countries of the European Union and the United States. This activity has also resulted in the development of medical tourism for phage therapy, which is often the last life-saving therapeutic option for many people [48,49,50,51].

Currently, studies on phages are the subject of research in many scientific centers around the world. Their main goal is the treatment and prevention of infections caused by MDR bacterial strains.

## 3. Phages as a Platform for Developing Vaccines

Phage-based vaccines have unique biological properties that make them ideal for the prevention and treatment of troublesome, recurrent, and chronic infections caused by various bacterial strains (including those resistant to antibiotics), viruses, fungi, and parasites and also for battling cancer (by eliciting an anti-cancer host immune response). Since phage vaccines are chemically stable, easy to store and transport, cheap to produce on a large scale, and capable of inducing a humoral and cell-mediated host immune response, they can be a valuable alternative to overcoming the limitations of conventional vaccines [18,21,22,52,53,54,55,56,57,58].

Phage vaccines can be of three main types, i.e., phage DNA vaccines, phage-displayed vaccines, and hybrid vaccines. Phage DNA vaccines are prepared by incorporating the eukaryotic expression cassette with a vaccine gene encoding an antigen or mimicking an epitope into the phage genome. Phage-displayed vaccines represent recombinant phages that display antigenic peptides or proteins on their surface as a result of genetic fusion with phage coat proteins. The third type of phage vaccines are hybrid vaccines, which are the result of combining the two types mentioned above, i.e., phage-displayed vaccines and phage DNA vaccines [18,19,55,56,57,58].

### 3.1. Phage DNA Vaccines: Design and Novel Applications

DNA vaccines are based on the administration of a plasmid DNA encoding an antigenic protein or peptide driven from the selected pathogen to stimulate the host’s humoral and cellular immune responses (mainly the cytotoxic T lymphocyte (CTL) response as a result of intracellular antigen expression). These vaccines ensure that the antigen is properly folded inside the host cells. Moreover, such target DNA can be obtained quickly and in large quantities, which significantly lowers the overall production costs. There is also no risk that the DNA vaccine will be pathogenic to the immunized organism [55,56,57,58]. However, DNA-based vaccines are not always effective due to their low stability and poor immunogenicity, especially when vaccinating large animals. Naked DNA molecules can be degraded, requiring various methods to extend the time of transgene expression and increase the efficiency of transfection by transporting DNA to target cells (e.g., electroporation, biolistic delivery based on DNA–gold micronanoplexes, lipid-based nanoparticles (LBNs)). In addition, optimization of the immunogenicity of such vaccines is also needed based on, e.g., codon optimization, genetic adjuvants, and sophisticated booster-priming schemes [56,59,60,61]. Nowadays, no commercial DNA vaccines are also known to be used for the prevention of infectious diseases in humans, but due to the severe acute respiratory syndrome coronavirus 2 (SARS-CoV-2) pandemic, intensive work is underway on potential vaccines (to limit the global spread of infection). A chimpanzee adenovirus-vectored vaccine (ChAdOx1 nCoV-19) expressing the SARS-CoV-2 spike protein passed the phase 1/2, single-blind, randomized controlled trial (RCT) at five UK research centers and showed an acceptable safety profile and the induction of both humoral and cellular immune responses [62].

Phage DNA vaccines with adjuvant properties can be an alternative eliminating the disadvantages of conventional DNA and phage-displayed vaccines, which in the case of some antigens do not allow their correct folding (compared to native primary conformations) or eukaryotic posttranslational glycosylation. This technology is based on the cloning of inserts encoding therapeutic antigens or epitopes in the eukaryotic expression cassettes under the control of specific promoters and regulatory sequences (required for gene expression and protein folding) incorporated into phage genomes and then in vitro packed into recombinant phage particles. This kind of vaccines are propagated in host bacterial cells. In vivo administration of recombinant phages results in their capture by antigen-presenting cells (APCs) and other immune cells, followed by lysis of phage carriers, release of genetic material, expression of protective antigens, and stimulation of the immune responses [18,19].

The most common phage vectors used as carriers for DNA vaccination are those based on the λ phage. In one study dependent on the λ strategy, the genes encoding green fluorescent protein (GFP) and hepatitis B small surface antigen (HBsAg) (under the control of the cytomegalovirus promoter (pCMV)) were cloned into λ-gt11 DNA to construct potential λ vaccines for nucleic acid immunization. Then, after mice were vaccinated, the immunogenic properties of recombinant λ-HBsAg phages (10^11^ plaque-forming units (PFU)) were tested in relation to naked DNA (pCMV–HBs plasmid DNA), unmodified λ-gt11, and HBsAg protein (control mice). It was shown that λ-HBsAg induced antibody responses (anti-HBsAg) in excess of 150 mIU mL^−1^, which were measured against recombinant HBsAg. The measured HBsAg antibody responses were significantly higher compared to the control mice. In turn, reporter phage λ-GFP was used to show the quick in vitro uptake of phage particles and expression of the target gene (inserted in the phage genome) by activated mouse peritoneal macrophages (as kind of APCs). The results obtained demonstrated GFP antigen on the macrophage surface 8h after incubation with whole recombinant λ particles (carrying the reporter gene), which underlines the high potential of the λ phage as a vaccine delivery vehicle. These analyses also indicate the possibility of direct immune stimulation by APCs in mice immunized with a phage DNA vaccine [63].

Other investigations using mouse and rabbit models also demonstrated the immunoprotective efficacy of whole λ phage particles carrying enhanced green fluorescent protein (λ-EGFP) or HBsAg (λ-HBsAg). After the third vaccination with λ-HBsAg, all four tested rabbits showed similar high-level anti-HbsAg responses that did not decrease after more than 6 months. The immune response in rabbits was higher compared to mice vaccinated with equivalent doses of λ-HBsAg, suggesting a swamping effect in the mouse model [64]. Subsequent work also based on the λ DNA vaccine expressing HBsAg revealed a strong antibody response in rabbits (after three vaccinations at 0, 5, and 10 weeks) that was higher compared to the commercially available Engerix B DNA vaccine consisting of the recombinant HBsAg protein (rHBsAg) [65]. The presented studies show the great potential of phage vaccines against infections caused by HBV (Hepatitis B Virus). This may be due to the tenfold-larger λ DNA (~50 kb) compared to the plasmid vectors (used for immunization of naked DNA) and thus the application of a 10 times’ lower dose of the λ DNA vaccine in the form of phage particles (per gene copy).

Another example of such research is recombinant λ phage nanoparticles encoding the core antigen of HCV (Hepatitis C Virus) as an alternative vaccine against infections caused by hepatitis C virus, which are a huge problem all over the world. The examination was based on a primary-boost system with naked DNA (pcDNA3-Core)-encoding core antigen and the recombinant λ-HCV phage. The studies showed that homologous prime/boost with λ-HCV stimulated higher levels of cellular and humoral immune responses in vaccinated mice in comparison to the DNA vaccine. Moreover, heterologous priming with DNA followed by boosting with λ nanoparticles induced highest level of core-specific immune responses in test animals [66]. This is an excellent approach that can be used to prepare a safe, novel vaccine strengthening the host’s immune system.

The presented strategy based on the λ phage shows many beneficial features, such as increased stability of target gene antigens, their capture by APCs and protection against degradation, high capacity for cloned inserts (up to 15–20 kb), no antibiotic-resistance genes, potential oral delivery, adjuvant properties, inherent biological safety for eukaryotic cells, and low vaccine production costs with high throughput [65,66].

Beside the λ phage system, recombinant filamentous phages are also used in the design of phage DNA vaccines. A good attempt to test this strategy is the recombinant M13 phage containing in its genome the expression cassette of glycoprotein D of herpes simplex virus type 1 (HSV-1) (crucial for viral adherence to host cells). Immunization of mice with whole particles of the recombinant filamentous phage induced host humoral and cellular immune responses, which underlines its usefulness in the construction of potential phage vaccines. In addition, filamentous phages can be an excellent alternative to λ-based vaccines due to their easier purification and production [67].

Phage DNA vaccines are generally economical (by phage propagation in prokaryotes) and safer than classical DNA vaccines due to the inability of phages to multiply in eukaryotic cells and the lack of pathogenic effect in the human body [63,64,65].

### 3.2. Phage-Displayed System, Vaccines, Construction, and Potential Applications

The technology of phage display was invented by G. Smith to display proteins and peptides on the surface of filamentous phages [37]. Currently, it is one of the most common phage systems used to produce large amounts of peptides, proteins, and antibodies [68]. The best-known phage display systems are based on the non-lytic M13 phage and the related filamentous phages belonging to the Ff class. Besides M13, this class also includes fd, and f1 phages infecting Gram-negative bacteria. The filamentous viruses contain a circular single-stranded DNA (cssDNA) genome surrounded by the virion capsid composed of the major coat protein, pVIII, which has a high copy number (2750 copies per phage nanofiber), and four minor coat proteins (five copies of each protein) at the two ends of the virion (pIII and pVI at the proximal end and pVII and pIX at the distal end, respectively). In addition, the non-lytic replication mechanism of M13 requires the secretion of all phage components via the inner membrane of the host cell before assembling into nanofibers. This may limit the exposed antigens in their sequence, length, and folding [69,70]. The phage display technology involves cloning a gene encoding a foreign protein/peptide to a specific site in the phage gene encoding one of the phage coat proteins (i.e., major or minor) and the production of a coat fusion protein with an exogenous sequence when infecting eukaryotic cells. Such a system offers the possibility of exposing the foreign antigen based on each copy of the target capsid protein as well as generating mosaic viruses in which the given capsid proteins present a mix of wild-type and recombinant proteins with the desired oligopeptide [18,19,71].

Phage display technology is also widely used for the construction of phage-displayed random peptide libraries to identify peptides as potential vaccine ingredients (selected by biopanning strategy) with great specificity and affinity to any desired (target) molecule. One study demonstrated the application of the M13 phage library as a promising approach for vaccine development based on selected immunogens mimicking antigens of *Rhipicephalus* (*Boophilus*) *microplus* ticks (infecting farm animals). Nine phage-displayed antigen peptides were validated and used to vaccinate mice and cattle (with various preparations, i.e., single peptides or a mixture of all peptides, respectively). Each of the phage clones elicited specific immune responses in the immunized animals, showing their potential role as a vaccine against ectoparasites [18,72]. On the other hand, the SARS-CoV-2 pandemic forced the rapid development of potential forms of therapy and the prevention of infections caused by this virus. Therefore, based on the phage-displayed Fab, scFv, and VH libraries, monoclonal antibodies (mAbs) directed against the angiotensin-converting enzyme 2 (ACE2) receptor-binding domain (RBD) of the spike protein of SARS-CoV-2, were quickly identified (within six days) by biopanning [73,74]. One selected mAb, IgG1 ab1, with high affinity specifically neutralized live SARS-CoV-2 in transgenic mice expressing human ACE2. This indicates the important role of large phage libraries in responding to public health threats in the face of the constantly emerging new microorganisms [74].

Phage-displayed vaccines are a particular example of the exposing strategy described above. These vaccines are recombinant phages that display guest antigens on their surface by genetic fusion with a selected coat protein. In this system, the proteins pIII and pVIII of filamentous phages are most commonly used for surface display of protective immunogens. The above phage proteins have different copy numbers, which affect the exposure level of the target antigens and the immunogenicity of the vaccine itself. In addition, these phage molecules can be employed as carriers of peptides or proteins of a certain size. Due to these limitations, the pVIII protein is used for the insertion of short peptides below nine amino acid residues in length, and pIII for the exposition of large proteins, approximately 100 kDa [37,75].

The DNA fragment encoding the vaccine peptide is cloned at the 5‘ end of the pVIII gene in the phage genome (displaying the peptide on each copy of the phage protein). Due to the fact that peptides longer than nine amino acids may interfere with the functions of the pVIII protein and the ability of recombinant phage particles to infect bacterial cells, a procedure for obtaining hybrid phages has been developed [76,77,78]. In such a hybrid system, the target peptide is exposed only on the pVIII copy fraction. In this way, the size of the foreign peptide can be increased to 14 or even 20 amino acids. Such an approach is based on a phagemid (harboring a viral gene encoding a coat protein, a phage origin of replication, and a phage packaging signal) and a helper phage that provides the wild-type coat proteins and phage assembly proteins but does not contain a phage packaging signal (e.g., a M13 phage derivative). As a result, recombinant phages contain mainly phagemid DNA due to the lower packaging efficiency of the helper phage compared to the phagemid [76]. On the other hand, when generating N-terminal fusion with the pIII protein, it is necessary to remove the natural stop codon and the 3‘ untranslated region at the end of the cDNA. The resulting cDNA libraries are often fragmented prior to cloning. The idea is to separate functional domains from ballast sequences [79]. Unfortunately, this system does not always work, and the selected cDNA clones are either non-functional or contain premature stop codons. A solution to this problem can be the procedure of selecting open reading frames before cloning them into phage display vectors [80]. In the above procedure, for the cloning of fragmented cDNAs, a special vector was used that conditioned the flanking of the target sequence by the pIII leader sequence at the 5′ end and the β-lactamase gene at the 3′ end (additionally flanked by the loxP sequences recognized by Cre recombinase). Only clones inserted into the reading frame produced β-lactamase, which conferred their resistance to ampicillin on agar medium. After the β-lactamase gene was excised, a direct ORF (Open Reading Frame) fusion with the pIII gene was obtained [80,81]

An interesting example of the design of this type of phage-based vaccines is also the recombinant M13 phage exposing on its surface three antigens, KETc1, KETc2, and GK1, and a recombinant porcine cysticercosis antigen, KETc7 (at multiple copies by the fusion to pVIII protein). Cysticercosis is a parasite disease in humans and rustic pigs caused by an armed tapeworm, *Taenia solium*. The oral administration of this phage vaccine in pigs is associated with the induction of a cellular response specific to target antigens. This is the first report demonstrating the use of the filamentous phage as a target vaccine in pigs (belonging to the group of large animals), which are the only intermediate hosts of the *T. solium* parasite [82]. Similar studies were also performed by another research group, administering the recombinant M13 phage that displayed on its surface one of the antigens—S3Pvac (composed of several protective peptides) of *T. solium*. Then, the anti-cysticercosis tripeptide vaccine (S3Pvac) was used to immunize pigs (in central Mexico). Vaccination resulted in a 70% reduction in tongue systeriocosis and a 54% reduction in musclecysticercosis, respectively [83].

Recently, the YGKDVKDLFDYAQE epitope from fructose bisphosphate aldolase was displayed on the coat proteins, pIII or pVIII, of the filamentous phage fd as an alternative vaccine against systemic candidiasis caused by *Candida albicans* strains. The genetically modified phages, 3-F and 8-F, were used to immunize mice, which were then infected with a non-lethal or lethal dose of *C. albicans* strains, respectively. Both recombinant phages, in addition to inducing humoral and cellular responses in the mice tested, extended their lifetime, reduced kidney damage, and decreased fungal loading in mouse kidneys [84].

Another research also based on the phage display strategy resulted in the generation of phage–peptide constructs that were able to elicit the production of antibodies against gonadotropin releasing hormone (GnRH). Since the phage vector had one copy of gene 8, all copies of the major coat protein VIII were modified by the fusion peptides. The specific peptides were selected from a phage display library using several types of GnRH antibodies, and five phage constructs were used to immunize mice (5 × 10^11^ virions). The induction of GnRH-specific antibodies was then observed in test animals without lowering serum testosterone levels, which is an indirect indicator of the neutralizing properties of GnRH antibodies. In contrast, one of the phage constructs after immunization of mice (with a dose of 2 × 10^12^ virions per mouse) in combination with poly(lactide-co-glycolide) (PLGA) adjuvant was responsible for a multiple increase in the production of GnRH antibodies and a significant reduction in serum testosterone levels. The studies confirmed that the antibodies generated in response to phage-GnRH immunization had neutralizing properties [85]. Further research of the above phage constructs revealed their contraceptive properties in vaccinated (sexually mature male) cats. After immunization, all cats produced anti-GnRH antibodies. Additionally, serum testosterone was suppressed 8 months after primary vaccination. However, more examinations are needed to identify the groups of animals most sensitive to this type of vaccine and to optimize the vaccination program [86].

Besides filamentous phages, which are the most common vectors of the display system, T4 and T7 lytic phages can be part of the phage display strategy for vaccine development. The capsid of the T4 phage is composed of 9-19 different proteins, among which are 2 non-essential capsid proteins, HOC (i.e., the smaller capsid protein) and SOC (i.e., the highly antigenic outer capsid protein), which can be used as the carriers of foreign peptides and proteins. Each phage particle contains 155-160 copies of HOC protein and 810-960 copies of SOC protein. Therefore, large immunogens can be expressed on the surface of phages in a high number of copies even more efficiently than using filamentous phages. The capsid proteins also have immunogenic properties and are able to elicit immune responses in humans and mice [87,88,89,90]. In addition to high immunogenicity, the T4 phage display system is characterized by the absence of secreted toxic proteins, phage assembly in the cytoplasm of host cells, and the possibility of dual exposure of foreign antigens on both SOC and HOC sites [89]. The T7 phage display system uses two forms of capsid proteins, 10A (415 copies) and 10B (1 copy), for the display of proteins or peptides. The 10B form is obtained by a translational frame shift at amino acid 341 of 10A. The system allows for the insertion of genes above 2 kb (high cloning capacity), the exposure of antigens in high copy numbers, and the induction of humoral and cellular responses in animal models [91,92].

One of the research groups obtained a high-density display system of single (p24-gag) or multiple (p24-gag, Nef, and gp41C-trimer) human immunodeficiency virus (HIV) antigens on the capsid surface of the reconstructed T4 phage. The p24 antigen (as a significant vaccine target) was used due to its high degree of sequence conservation among various HIV isolates. A potential anti-HIV vaccine was obtained by fusing the gene encoding the HIV antigen to the Hoc gene (5′ or 3′ end), followed by the expression and production of purified antigens and then the in vitro assembly of *hoc*^-^ (Hoc-deficient) phage particles via Hoc–capsid interactions. Based on the novel approach, it was possible to attach more than one fusion HIV antigen (using a single binding step) on the same phage capsid (without inserting the HIV genome into the immunogen) in order to construct a multi-component vaccine. Moreover, the displayed p24 were able to induce strong humoral and cellular immune responses in mice, indicating the great potential of the phage T4 system as a novel vaccine against HIV [93]. In other studies, the effectiveness of the recombinant T7-GH phage displaying (on its surface) the G-H loop region (the major neutralizing antigenic site) of the VP1 structural protein of foot-and mouth disease virus (FMDV) was demonstrated on the basis of a pig model. FMD is a highly contagious infection that affects cloven-hoofed animals, causing serious economic losses around the world. Pigs immunized with the recombinant phages induced stronger antigen specific immune responses than the commercially available PepVac vaccines, indicating the enormous importance of these studies in the development of an FMDV phage-based vaccine [94].

Additionally, the λ phage is also used as a platform for phage display of peptides and proteins. Compared to filamentous phages (e.g., M13), the recombinant λ phages allow the display of multiple copies of two to three times larger fusion proteins without disturbing the phage morphology. The display level of fusion proteins on the D head outer capsid protein (in 420 copies) particles is also higher. Moreover, the λ system can be employed for the presentation of peptides, in the case of which membrane secretion (systems based on filamentous phages) is difficult [18,95,96,97]. In addition, the λ phage is used to a great extent as a DNA vaccine vector [66,98,99].

Subsequent studies demonstrated the possibility of using the λ phage for high-density display of peptides and proteins (exposed in high numbers on the phage surface), which leads to their efficient selection in biopanning. The system also included proteins containing multiple disulfide bridges, fused to the C-terminus of the λ D protein (Cys-display system). Compared to the widely used M13 phage as a gene carrier, the λ display system was several orders of magnitude more efficient. Due to the large size of the λ genome, the target DNA sequences were cloned into a donor plasmid vector (pVCDcDL1) containing the gene sequence encoding the λ D protein and the lox sequences (lox Pwt and lox P511) necessary for recombination and integration of the plasmid DNA into λ (lDL1) DNA. After transformation of *E. coli* cells expressing Cre recombinase with the donor plasmid and infection with lDL1 (containing the lox Pwt and lox P511 sequences), recombination occurred at consistent lox sites in both vectors. The high-density λ system has also been successfully used to map the epitopes of monoclonal antibodies and human polyclonal serum and toxin molecules based on a library of toxin gene fragments displayed on λ particles. Single-chain fragments (scFV) of the antibodies were exposed in a functional form on the phage surface, which indicated the correct formation of disulfide bonds in the proteins studied [100].

The application of the λ system was perfectly confirmed by the research related to the display of four immunodominant regions of porcine circovirus 2 (PCV2) capsid protein (LDP-D-CAP) on the surface of recombinant phage particles. The immunogenic antigen (containing the sleeted epitopes) was obtained by the fusion to the carboxyl-terminal of D-capsid protein. Vaccination of pigs with LDP-D-CAP (without adjuvant) induced humoral and cellular immune responses against PCV2. Furthermore, no adverse local and systemic reactions were observed in test animals as an effect after administration of the λ vaccine [101,102].

Recombinant λ phages have also shown efficacy as mucosal vaccine carriers for selected disease-specific epitopes (DSE) derived from the cervid prion protein (amino acids 130-140 [YML]; 163-170 [YRR]; and 171-178 [YRR]) in fusion with the λ D-capsid protein (LDP-DSE). The λ vaccine, administered to mice and newborn calves, was distributed in the gastrointestinal tract 24 h after oral dosing. Target delivery of purified LDP-DSE to the small intestine resulted in the induction of different IgA responses to all three antigenic peptides. In addition, delivery of bacteria expressing soluble D-DSE also stimulated epitope-specific IgA responses in the target Peyer’s patches. These studies were the first to demonstrate the use of LDP to induce epitope-specific IgA responses in the small intestine. Moreover, the examinations highlighted the function of Peyer patches as a site of λ phage capture [103].

Other studies have reported fusions of human and porcine cathelicidines LL37 and PR39, defensins HBD3 and DEFB126-Δ (deleted for its glycosylation sites), and HD5 defensin to the λ D-capsid protein (including both NH_2_ and COOH ends). In the case of COOH-terminal fusions, a toxic effect was observed in *E. coli* cells compared to proteins fused to NH_2_ end of the λ D-capsid protein. The toxicity of the peptides fused to the COOH terminus may result from the oxidation of defensins, leading to the formation of three intramolecular disulfide bonds within the cytoplasm of *E. coli* cells and obtaining a biologically active (highly toxic) conformation by the proteins. Moreover, both cathelicidin and α- and β-defensins fused with the λ D-capsid protein showed antibacterial activity. The described λ system based on high-density single epitope presentation indicates its usefulness as a potential vaccine against various infectious diseases [104]

The above research results indicate that phage display technology has enormous potential for use in a wide range of biomedical and pharmaceutical fields (e.g., construction of random peptide libraries, vaccine development, analysis of protein–protein interactions, or isolation of disease-specific biomarkers) [18,19,54,68].

### 3.3. Hybrid Vaccines

Another example of the use of phages in the preparation of immunogenic formulations is hybrid dual vaccines that combine phage-displayed and phage DNA vaccines. Therefore, these vaccines, by displaying a protein or peptide with high affinity to APCs and through the insertion of an eukaryotic expression cassette encoding a specific antigen into the phage genome (dual effect), result in enhanced stimulation of host immune responses [19,105]. The main purpose of such preparations is to increase the interaction of phages with APCs and then internalization of bacterial viruses by these cells. Moreover, vaccines of this type are an excellent strategy for the targeted delivery of antigenic genes (via an unknown mechanism) to dendritic cells (DCs), which function as the major APC population critical in triggering a cytotoxic anti-tumor response, based on CTLs [52,95,106,107].

An excellent model of this vaccine system are genetically modified filamentous phages of fd virions, which greatly enhanced the immunogenicity of tumor-associated antigen-derived peptides (i.e., HLA-A2-restricted MAGE-A10_254–262_ or MAGE-A3_271–279_) and could be used as a potential anti-cancer vaccines. The phage preparations induced strong anti-tumor CTL responses in vitro and in vivo [107]. In addition, the CTLs specific to the MAGE-A3_271-279_ antigen played a significant role in the active killing of human MAGE-A3^+^ tumor cells. Furthermore, an in vivo tumor protection assay revealed that vaccination of transgenic humanized HLA-A2.1^+^/H2-D^b+^mice with fd phage particles, expressing and delivering peptides derived from MAGE-A3271-279, inhibited tumor growth [95]. The results obtained indicate that the above double-hybrid filamentous phage fd system is ideal for the production of peptide-based anti-cancer vaccines [107].

The presented research results indicate the great utility of phage-based vaccines in preventing infectious diseases in humans (Table 2).

## 4. Personalized Medicine and Clinical Trials Based on Phages (Determining the Safety of Phage Therapy)

The personalized medicine model is closely related to the development of a large bank of phages that constitute active pharmaceutical ingredients (APIs) for fighting bacterial infections. To administer this form of therapy to a patient, it is necessary to isolate and define a clinical pathogenic strain and then select from the bank of an active phage directed against it. In the next step, a phage medicinal product unique for a given patient is obtained. Works of this type are carried out in Europe and the United States [35]. For some patients, monophage therapy based on one type of phage (characterized by a narrow host range) is highly effective. Unfortunately, such therapy sometimes carries the risk of a given bacterial strain developing resistance to the selected type of phage. Moreover, during long-term therapy, the treatment can elicit host immune responses and induce the production of anti-phage antibodies. Accordingly, phage therapy based on the use of two or more phages or phage cocktails (i.e., polyphage therapy) is sometimes more effective than monophage therapy. This conditions a wider host spectrum and avoids the undesirable effects that may occur with monophage therapy. Therefore, polyphage therapy is effective primarily in the case of infections caused by different strains of bacteria (strategy based on the recognition of different receptors by phages) or diseases related to the formation of biofilm [108,109].

Phage therapy is currently the subject of research in many scientific and medical centers around the world, especially in the case of multidrug-resistant infections in humans. Since 2007, phage therapy has been allowed for the treatment of bacterial diseases at the Queen Astrid Military Hospital (QAMH) in Brussels, Belgium, under the auspices of the Declaration of Helsinki (Article §37) established by the World Medical Association. The external demand for phage therapy at this hospital has increased significantly in the past few years. In 2013–2018, 260 patient requests for phage treatment were received, of which only 15 cases were qualified for this form of therapy [110]. In addition, in January 2018, the Belgian federal government developed legislation on phage production and clinical use of therapy based on phages. In Belgium, a novel approach, known as the magistral phage preparation (or in the U.S., compound prescription drug preparation), has also been introduced, on the basis of which, following the doctor’s recommendations, non-standard phages (intended for use in personalized therapy) are obtained in the laboratory [111]. Similar phage-based therapeutic activities (known as experimental therapy) are carried out at HIIET in Poland [48,51].

Extensive phage-based clinical research is needed to spread this form of therapy (as medicinal preparations) in the treatment of incurable infections. Therefore, case reports of the clinical use of phage therapy are so important. One of the PhagoBurn clinical trials assessed the safety and efficacy of using phages in human therapy. PhagoBurn is a multicentered European research and development project (NCT02116010) that was funded by the European Commission in 2013. The clinical trials for this project were ended in 2017. The aim was to evaluate the potential of phages in the treatment of burn wounds infected with MDR *E. coli* and *P. aeruginosa* [112]. Chronic wound management associated with bacterial infections and biofilm formation is a huge problem for healthcare systems around the world. These types of wounds still result in high rates of limb amputation and premature death. For this reason, alternative forms of treatment are needed to fight drug-resistant bacteria [12,113]. As part of the PhagoBurn project, a randomized, controlled, double-blind phase 1/2 study (European Clinical Trials database, number 2014-000714-65) was performed. The clinical trial was based on a phage cocktail containing 12 phages with lytic activity against *P. aeruginosa* (PP1131, 1 × 10^6^ PFU/mL) and included 26 patients with burn wound infections (from nine burn centers at French and Belgian hospitals) and 1 patient without infection. Topical phage therapy lasted for 7 days and, wounds were followed for an additional 14 days. The control group (13 patients) received standard care treatment, including 1% silver sulfadiazine emulsion cream applied topically. Insufficient efficacy of the phage preparation, resulting from the use of lower-than-intended phage concentrations (a daily dose of approximately 10–100 PFU/mL instead of the expected dose of 1 × 10⁶ PFU/mL), led to the termination of the study. In addition, it was shown that bacteria isolated from patients undergoing unsuccessful treatment of PP1131 were resistant to low doses of phages. However, the control group was characterized by a greater number of adverse effects compared to phage therapy, such as bronchitis, pneumonia, and septic shock [112].

Other studies have looked at the use of a cocktail of customized bacteriophages in the treatment of chronic non-healing wounds infected with *Escherichia coli*, *Staphylococcus aureus*, and *Pseudomonas aeruginosa*. The examinations involved 20 patients (12–60 years old) who did not respond to conventional local debridement and antibiotic therapy. After three to five doses of topical treatment with phages, no signs of microbial and clinical infection were found. Seven patients healed completely after 21 days of phage therapy, while the others had healthy margins and healthy granulation tissue. This demonstrates the effectiveness of topical bacteriophages in the treatment of chronic, non-healing wounds [12].

A good example of personalized medicine is the use of intravenous phage therapy to treat a 26-year-old patient with CF suffering from MDR *Pseudomonas aeruginosa* pneumonia, persistent respiratory failure, and colistin-induced renal failure. CF patients, due to continuous exposure to antibiotics, are more prone to infections caused by MDR bacteria, especially *P. aeruginosa*. The respiratory colonization by these MDR strains increases with the age of the patients. Consequently, chronic respiratory infections are the underlying cause of the morbidity, progressive respiratory failure, and mortality in these patients. Phage therapy in such cases can be a life-saving medical procedure. In the described case, no recurrence of the pseudomonal pneumonia was observed within 100 days after the end of phage-based therapy. Moreover, in the test patient, there was no exacerbation of CF. This determined the performance of a bilateral lung transplant in this patient after 9 months, emphasizing the usefulness of phage therapy in this group of CF patients [13].

The widespread administration of antibiotics has also led to the development of opportunistic strains of *Acinetobacter baumanii* that are resistant to first-line treatment and last-resort antibiotic therapy, hence the need to develop alternative forms of therapy for this type of infections. A perfect example of such therapeutic activities against clinical isolates of *A. baumannii* is the phage vB_AbaM_PhT2 (collected from hospital wastewater in Thailand). Its efficacy has been proven in the treatment of human brain and bladder cell lines grown in the presence of *A. baumannii*. The cells released significantly less lactate dehydrogenase compared with cells not treated with this phage. This confirmed the efficacy of phage and its potential application as a surface antimicrobial agent for use in hospitals [114]. Another study used a novel approach to prepare personalized therapeutic phage cocktails against the life-threatening MDR infection of *A. baumannii* in a 68-year-old diabetic patient with necrotic pancreatitis. The repeatedly applied antibiotic treatment did not improve the patient’s health. In the absence of an effective therapy, two laboratories isolated the *A. baumannii* strain from the patient, and then nine phages with specific lytic activity against this pathogen were selected (to make phage cocktails as an emergency investigational new drug (eIND), approved by the Food and Drug Administration (FDA)). Intravenous and transdermal administration of the phage preparation to the patient resulted in elimination of the infection and the patient’s recovery [15].

The rapid emergence and spread of uropathogenic *E. coli* strains (UPECs) that are resistant to widely used antibiotics is also a serious medical problem. These difficulties create the need to develop innovative forms of treatment, such as phage therapy [115,116]. A significant approach to clinical trials involving uropathogenic strains (e.g., *Staphylococcus aureus*, *E. coli*, *Streptococcus* spp., *Pseudomonas aeruginosa*, *Proteus mirabilis*) was based on commercially available pyophage (Pyo) (Eliava BioPreparations Ltd., Tbilisi, Georgia). The treatment procedure (preceded by the project of a prospective, randomized, placebo-controlled, double-blind clinical trial) was successful in six of nine patients, with transurethral resection of the prostate (~67%). In these persons, a reduction in bacterial count was shown without any adverse effects, emphasizing the effectiveness of this phage-based therapy [14]. UTIs associated with catheterization (CAUTIs) are also a huge therapeutic challenge. This is due to the formation of biofilm by uropathogens and the inhibition of the action of antibiotics in the urinary tract. One of the preclinical studies (made on the dynamic biofilm model simulating CAUTIs) indicates an important role of phages in the reduction of biofilm formation by *Proteus mirabilis*, even during 168 h of catheterization. This shows the importance of the applied phage therapy (based on the phage cocktail composed of the podovirus vB_PmiP_5460 and the myovirus vB_PmiM_5461) both in the treatment and prevention of CAUTIs and in the development of bacterial biofilm on catheters and other similar medical devices [117].

The analysis of clinical data shows the high effectiveness of phage therapy as an alternative therapeutic strategy against MDR infections in humans. On the other hand, the use of phages in patients after solid organ transplantation has been poorly studied. One of the reported phage-based therapies involved three lung transplant recipients (LTRs) suffering from life-threatening infections caused by *Pseudomonas aeruginosa* (two patients) and *Burkholderia dosa* (one patient) [16]. On the basis of the isolated bacterial strains, lytic bacteriophages were selected for therapy. Simultaneously with the phages, the patients were given antibiotics. In the above-mentioned persons, no adverse symptoms were found in connection with the phage therapy used. The treatment was well tolerated and resulted in the observed clinical improvement. Two patients (ventilator dependent) infected with resistant *P. aeruginosa* responded clinically to phage therapy and were discharged from the hospital without a ventilator. In the case of the third patient, suffering from recurrent *B. dolsa* infections, his clinical condition improved after initiating phage treatment and he was disconnected from the ventilator. Unfortunately, the infection returned during phage treatment and the patient died. In this person also, no adverse events were observed resulting from the use of phage-based therapy.

*Mycobacterium tuberculosis* infections are also a serious global threat. The resistance to tuberculous and non-tuberculous mycobacteria (NTM) is widespread [118]. In CF patients, NTM infections are particularly persistent, which can lead to their death after lung transplantation. Phage therapy may be a solution to this problem, along with reducing morbidity and mortality from NTM. The efficacy of phage activity was confirmed by its use in CF patients with disseminated *Mycobacterium abscessus* infection, after bilateral lung transplantation. Intravenous treatment with a cocktail of three phages (obtained as a result of genetic engineering) was applied, which resulted in the improvement of the patients’ health, closure of the sternum wound, improvement of liver function, and reduction in infected skin nodules [17].

The presented examples of clinical trials and the applied personalized phage therapies indicate their high efficiency and safety. However, more clinical trials are still needed on a larger number of patients to establish this form of therapy as a routine treatment procedure, similar to antibiotics today.

## 5. Conclusions

Currently, phage therapy is of great interest to scientists and physicians, especially due to successful attempts to treat patients for whom traditional antibiotic therapy has failed. Moreover, phage therapy represents a renewed approach to fight MDR bacterial strains [35,111]. Phage therapy can be also individually selected for each patient as personalized treatment. Bacteriophages are much more specific and safer than antibiotics because they do not have the ability to multiply in eukaryotic cells, and the applied therapy shows no side effects and does not adversely impact the host commensal (health-protecting) microflora [20,21]. These bacterial viruses also do not show pathogenic and harmful activity toward model animals or humans. An additional advantage of phages is the possibility of replication in vivo (autodosing system), which results in the possibility of using low dosage of this type of medicinal preparations compared to antibiotics [119,120]. Due to the relatively narrow host range displayed by most phages, the generation of phage resistance mechanisms by bacteria with respect to antibiotic resistance is also low [20,21,116].

Phage vaccines can provide potential protection for humanity against newly emerging pathogens. Vaccines can be prepared according to three strategies. The first, involving phage-displayed vaccines, is associated with the fusion of the antigen to the phage coat protein (phage display technology). The second determines the insertion of the gene encoding the immunogen into the phage genome (phage DNA vaccines). The third is based on the combination of phage-displayed and phage DNA vaccines to enhance the host’s immune responses. Vaccines of this type show adjuvant properties, they can be easily produced on a large scale, and the costs of their transport and storage are lower compared to conventional vaccines [18,19,22,107]. However, further research is needed on the immune mechanism of action of phage vaccines, their safety, the delivery system of heterologous antigens, and the implementation of higher doses of these vaccines compared to those currently studied.

Classical phage therapy, in the face of the increasing number of antibiotic-resistant bacterial strains, has become a valuable, alternative therapeutic tool with a wide range of applications. More and more scientific communities, doctors, and pharmacists are starting to notice the unique properties of phages and the real possibilities of their use in the treatment of various infectious diseases affecting people all over the world. This is mainly due to the clinical trials conducted and the spectacular results achieved in the treatment of patients for whom proven medical interventions have been ineffective or the available treatment options have been exhausted. Some countries, such as Belgium and Poland, introduced a treatment system based on prescribing phages as selected therapeutic agents [35,48,51,111]. Thus, personalized phage therapy (adapted to individual patients and the type of pathogens infecting them) has slowly begun to develop. In many cases, it is a form of treatment that allows an infected person to return to full health, and sometimes even life-saving therapy. Unfortunately, there are still no legal regulations that would allow the application of phages as conventional medicinal preparations, and not only as part of experimental therapy that requires the consent of the bioethics committee and the patient.

Overall, extensive research indicates that phages are safe for the human population, exhibit bactericidal activity, even in the case of antibiotic-resistant strains, and are an innovative form of therapy compared to commonly used antibiotics. However, phage therapy requires far-reaching regulations and more RCTs, to be considered a routine medical therapy not just a therapeutic experiment saving human life.

## Figures and Tables

**Table 1 pharmaceuticals-14-00331-t001:** Highlights in the history of phage research and the initial development of phage therapy [24,25,26,27,28,29,36,37,38,39,40,41].

Selected Aspects of the History of Phage Research		
Date	Discoverer/Founder	Discovery/Investigation/Event
1896	Ernst Hankin	Antibacterial activity of river water samples against *Vibrio cholerae*
1898	Nikolay Gamaleya	Antiseptic action of an unidentified substance against *Bacillus subtilis*
1915	Frederick Twort	Hypothesis about ultramicroscopic viruses as antibacterial agents
July–August 2015	Felix d’Herrelle	Studies on developing a vaccine against dysentery caused by *Shigella*
18 October 1916	Felix d’Herrelle	Development of the term bacteriophages (from the combination of two words, bacterium and phage)
1917	Felix d’Herrelle	Documented discovery of phages (meeting of the Academy of Sciences and publication of dysentery research)
1919	Felix d’Herrelle	The use of an antidysentric phage in the treatment of dysentery—the birth of phage therapy (the Hospital des Enfants-Malades, Paris, France)
1923	George Eliava	Foundation of the Eliava Institute of Bacteriophage, Microbiology, and Virology (EIBMV) of the Georgian Academy of Sciences (Tbilisi, Georgia)
1939	Helmut Ruska	The first electron microscopy of phages that lyse bacteria
1952	Ludwik Hirszfeld	Foundation of the Hirszfeld Institute of Immunology and Experimental Therapy (HIIET) of the Polish Academy of Sciences (Wroclaw, Poland)
1977	Frederick Sanger	Sequencing of the first phage genome
1985	George P. Smith	Inventing the technology of filamentous phage display
1988	George P. Smith	Phage display library construction
1988/1989	Richard Lerner and Sir Gregory P. Winter	Phage antibody production: filamentous phage displaying antibody variable domains
1988	VF de la Cruz	Invention of a model phage-displayed vaccine against malaria caused by *Plasmodium falciparum*
2012	Richard Lerner and Sir Gregory P. Winter	Prince of Asturias Award for technical and scientific research
2018	George P. Smith and Sir Gregory P. Winter	Nobel Prize in Chemistry for phage display technology (directed evolution of antibodies, with the aim of producing new pharmaceuticals)

**Table 2 pharmaceuticals-14-00331-t002:** Types of phage-based vaccines—brief characteristics [18,19,37,55,68,69,70,88,91,95,96,97,105,106].

Type of Vaccine	Characterization
Phage DNA vaccines	The antigen gene is cloned in a eukaryotic expression cassette (under the control of a specific promoter) within a phage genome. The inserted encoding antigen is in vitro packed into the recombinant phage particles. The λ phage is the most common phage vector (large gene capacity) for DNA vaccination, although filamentous phages (multiple gene copies using a single vector) are also useful. DNA is protected from degradation by the capsid of phage particles. The antigen is expressed and folded correctly inside the host’s eukaryotic cell. Vaccine production is economical and effective due to the multiplication of phages in bacteria. Phage vaccines are more stable for storage, transport, and administration (i.e., by oral route) compared to classical vaccines. Whole phage particles (as vehicles for protective genes) can elicit effective immune responses in large animals.
Phage-displayed vaccines	The antigen gene is genetically fused to one of the phage coat proteins by the cloning procedure. A fusion coat protein with a foreign gene is obtained when phage particles express their genome while infecting a eukaryotic host. Filamentous phages (M13, fd, and f1), lytic phages (T4 and T7), and the temperate λ phage are common vectors used in this system. The potential use of the system is for the production of antigen-displaying vectors or the construction of phage-displayed libraries to identify new antigens (by biopanning). Recombinant phages induce humoral and cellular responses in animal models.
Hybrid vaccines	A vaccine (as a combination of phage-displayed and phage DNA technology) is based on: - a phage displaying on its surface peptides with high affinity toward APCs - a DNA plasmid encoding the therapeutic antigen in a eukaryotic expression cassette Phage particles are subject to efficient cellular internalization, and the efficiency of nuclear uptake should be increased. Hybrid phages improves cellular and humoral immune responses. They have potential application in the fight against cancers.

## Data Availability

Data sharing not applicable.

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
