# Peer review of "Bacteriophages as Potential Tools for Use in Antimicrobial Therapy and Vaccine Development"

_pharmaceuticals, 2021, doi:10.3390/ph14040331_

Round 1

Reviewer 1 Report

Title: Bacteriophages as Potential Tools for Use in Antimicrobial Therapy and Vaccine Development

General comments

This manuscript presents an actual and interesting review about the use of phages in vaccine development and personalized medicine. The manuscript was well elaborated and well written. I therefore recommend publication after minor revisions

Specific comments

Authors should standardize the word "bacteriophage or phage".

Abstract must be revised in order to improve the information presented. At this moment, the authors present the problem and the solution, but do not mention what was expected when they decided to write this review. Why a review on this subject is necessary? What lack of knowledge the authors searched to fulfil with this research? This is the first review on this subject? The authors should consider restructuring the abstract with a finishing sentence describing the scope and aims of their review.

Lines 64-65 “Moreover, phage vaccines in the form of phage-displayed vaccines or phage DNA vaccines are currently of great interest among various research groups”. Please add references

Lines 192-193 “Phage vaccines can be used in two main types i.e. as phage DNA vaccines and phage-displayed vaccines”. The authors indicated that the vaccines can be used in two main types. However, in the following lines they describe three types of vaccines. I understand that the latter is the combination of the 2 types of vaccines, but it can mislead the reader.

Lines 429- 430 “The phage preparations induced strong anti-tumor CTL responses in vitro and in vivo”. Please add reference.

Lines 421 – 433 “Furthermore, an in vivo tumor protection assay revealed that vaccination of transgenic humanized HLA-A2.1+/H2-Db+mice with fd phage particles, expressing and delivering peptides derived from MAGE-A3271-279, inhibited tumor growth. Please add reference.

Lines 568- 595 The conclusion of the manuscript has many details that do not fit as a concluding statement. Please consider re-organization of the conclusion, highlighting the new research directions in the field the paper is addressing and future implications.

Author Response

I would like to thank the Reviewer for reading the manuscript, devoting time to analyzing the content, very accurate comments that will certainly improve the quality of the presented material.

Best regards,

Beata Zalewska-PiÄ…tek

Reviewer 2 Report

Zalewska-Piatek and coworkers present the use of bacteriophages as therapeutics for several disease states. Specifically, they demonstrate the use of bacteriophages as a replacement for traditional antibiotics and to address the increase in antibiotic resistant infections. Additionally, they demonstrate the use of bacteriophage particles in vaccine development as an antigen presentation platform and a delivery vehicle for DNA-based vaccines. The use of bacteriophage therapy is not novel and much work in the field has been produced. This review provides an updated source in advances in this area and describes some advances in the development of personalized phage therapies in the future.

Minor Concerns:

  • A review of the manuscript for English grammar is recommended to improve overall quality.
  • Lines 289-292 – the description of filamentous phage proteins is confusing in the current format. The reader may get the impression that only a single copy of the pVIII major coat protein is present.
  • Line 325-327 – the insertion size for proteins is reversed. The pIII protein is used for large insertions and the pVIII is used for the expression of smaller peptides <9 amino acids in length. A type 8+8 system would be required for display of longer peptides on the pVIII molecule.

Author Response

Thank you very much for reviewing the manuscript and attached comments.

Best regards,

Beata Zalewska-PiÄ…tek

Reviewer 3 Report

In the review “Bacteriophages as Potential Tools for Use in Antimicrobial Therapy and Vaccine Development” of Beata Zalewska-PiÄ…tek and RafaÅ‚ PiÄ…tek, the authors reported on pros and cons of phage application and therapy to treat bacterial infections.

The review is well written and the information is carefully summarized. This reviewer feels that the broad readership will be highly interested in reading this review. I have only minor things to improve the manuscript. Specific points that should be revised as specified below.

Tables 1 and 2 can be shifted to the Supplement! The information is nice to have, but not essential for the review.

Line 124: “Virusforschungan“ there is something wrong…

Author Response

We would like to thank the Reviewer for reviewing the manuscript, taking the time to read the entire material, and for valuable comments.

Best regards,

Beata Zalewska-PiÄ…tek
